

# Post-flight analysis of detailed size distributions of warm cloud droplets, as determined in situ by cloud and aerosol spectrometers

Sorin Nicolae Vâjâiac[1], Andreea Calcan[1], Robert Oscar David[3], Denisa-Elena Moacă[1,2],
Gabriela Iorga[2,4], Trude Storelvmo[3], Viorel Vulturescu[5], Valeriu Filip[1,6,7]

[1]National Institute for Aerospace Research "Elie Carafoli", Bucharest, Romania
[2]University of Bucharest, Faculty of Physics, P.O.Box MG-11, Magurele, 077125, Romania
[3]Department of Geosciences, University of Oslo, Oslo, Norway
[4]University of Bucharest, Faculty of Chemistry, Department of Physical Chemistry (Physics Group), Bd. Regina Elisabeta 4-12, 030018, Bucharest, Romania
[5]Politehnica University of Bucharest, Faculty of Industrial Engineering and Robotics, Theory of Mechanisms and Robots Department, Bucharest, Romania
[6]Graduate School of Physics, University of Bucharest, Romania
[7]Research Center for Surface Science and Nanotechnology, University POLITEHNICA of Bucharest, Romania

*Correspondence to*: Sorin Nicolae Vâjâiac (vajaiac.sorin@incas.ro) and Valeriu Filip (vfilip@gmail.com)

**Abstract.** Warm clouds, consisting of liquid cloud droplets, play an important role in modulating the amount of incoming solar radiation to Earth's surface and thus, the climate. The size and number concentration of these cloud droplets control the reflectance of the cloud, the formation of precipitation and ultimately, the lifetime of the cloud. Therefore, in situ observations of the number and diameter of cloud droplets are frequently performed with cloud and aerosol spectrometers, which determine the optical diameters of cloud particles (in the range of up to a few tens of microns) by measuring their forward scattering cross sections in visible light and comparing these values with Mie-theoretical computations. The use of such instruments must rely on a fast working scheme consisting of a limited pre-defined uneven grid of cross section values that corresponds to a theoretically derived uneven set of size intervals (bins). However, as more detailed structural analyses of warm clouds are needed to improve future climate projects, we present a new numerical post-flight methodology using recorded particle-by-particle sample files. The Mie formalism produces a complicated relationship between a particle's diameter and its forward scattering cross section. This relationship cannot be expressed in an analytically closed form and it should be numerically computed point by point, over a certain grid of diameter values. The optimal resolution required for constructing the diagram of this relationship is therefore analysed. Cloud particle statistics are further assessed using a fine grid of particle diameters in order to capture the finest details of the cloud particle size distributions. The possibility and the usefulness of using coarser size grids, with either uneven or equal intervals is also discussed. For coarse equidistant size grids, the general expressions of cloud microphysical parameters are calculated and the ensuing relative errors are discussed in detail. The proposed methodology is further applied to a subset of measured data and it is shown that the overall uncertainties in computing various cloud parameters are mainly driven by the measurement errors of the forward scattering cross section for each particle. Finally, the influence of the relatively large imprecision in the real and imaginary parts of the refractive index of cloud droplets on the size distributions and on the ensuing cloud parameters is analysed. It is concluded that, in the presence of high atmospheric loads of hydrophilic and light absorbing aerosols, such imprecisions may drastically affect the reliability of the cloud data obtained with cloud and aerosol spectrometers. Some complementary measurements for improving the quality of the cloud droplet size distributions obtained in post-flight analyses are suggested.



## 1. Introduction

Understanding the microphysics of clouds is a key component both in assessing future climate change and in operational weather forecast, with vast implications for modern domestic activities ranging from agriculture to energy harvesting and aviation. The cloud droplet size distribution has long been recognised as particularly important for the Earth's energy balance through the so-called cloud albedo effect (Twomey, 1977). The in-cloud microphysical processes involved in this effect are strongly influenced by the spatiotemporal variation in the detailed shape of the cloud droplet size distribution (see, for example, Feingold et al., 1997; Liu and Daum, 2002; Iorga and Stefan, 2007; Liu et al., 2008; Chen et al., 2016). It is currently recognized that in situ measurements are required to properly characterize the highly complex microphysical processes occurring in clouds in order to efficiently apply various models for resultant cloud albedo. In this context, as in situ investigations continue to offer the best spatiotemporal accuracy of cloud droplet measurements, one of the most useful types of airborne instruments is casted into the generic name of Cloud and Aerosol Spectrometer (CAS). Such devices, which are essentially a variant of the so-called optical particle counters (OPC), sort out cloud droplets based on their optical diameters, by measuring the forward scattering cross section (FWSCS) of a laser beam of known wavelength from cloud droplets entering the sample volume of the instrument (Baumgardner et al., 2001). The standard CAS measurement procedure can be split into two distinct phases, an instrumental and a numerical phase. The instrumental phase deals with a broad range of problems such as bringing the studied particles into the laser beam (within an air stream flowing with a known rate), selecting valid particles, collecting the scattered light on specialized sensors, and amplifying and recording the electrical output etc. The net product of this process is the measured value of the FWSCS for the qualified cloud particle.

Meanwhile, the numerical phase of a CAS measurement crucially involves the comparison of this measured value to the theoretical scattering cross section of pure water spheres (computed within the classical Mie formalism). The instrumental phase of the CAS measurement procedure is well documented in the literature (Baumgardner et al., 1985; Baumgardner and Spowart, 1990; Baumgardner et al., 1992; Baumgardner et al., 2001; Baumgardner and Korolev, 1997; Glen and Brooks, 2013) and will not be discussed in the present study. Instead, the focus will be on the numerical phase leading to the optical sizing of the cloud particles.

The typical range of particle diameters that can be analysed by a CAS is between 0.5-50 µm. However, the comparison step is often ambiguous due to the complicated quasi-monotonic dependence of the scattering cross section on the diameter of the target sphere. Owing to this behaviour, a measured value of the cross section corresponds in most cases to several diameters. Partly to alleviate this drawback, and partly to accelerate the (in-flight) comparison step, the size distribution is commonly constructed over a limited partition of uneven widths called bins. The limits of each size bin should be established unambiguously, in the sense that to each boundary corresponds a FWSCS value, or threshold, which cannot be assigned to any other diameter of a pure water sphere. The user has some freedom in setting the limits of the size bins, but the choice should be made in such a way that the corresponding thresholds of FWSCS are all unambiguous. The result is a partition of the FWSCS range in an equal number of uneven intervals, or cross section bins, associated to the chosen structure of the size bins. During the in-flight data acquisition, the measured values of FWSCS for qualifying cloud droplets are readily "sifted" through the grid of cross section bins and then assigned and counted in the suitable diameter bins. To optimize the statistical analysis of the cloud droplets, the operator should choose the limits of the diameter bins according to the range of droplet sizes expected in the sampled cloud. If, for example, the main focus is on small (few



microns) droplets, then more size bins should be designated in the range of such diameters. For reasons that will become clear in the next sections, the number of bin limits having unambiguous FWSCS thresholds tends to be larger in this case. However, for counting mainly droplets that are larger than 10 μm the dependence of the FWSCS on the particles' diameters becomes so riddled that fewer size bins with valid thresholds can be assigned. With

wider bins, the ensuing size distributions obviously become less accurate. The sizing precision can be improved if each particle's FWSCS response is considered separately and its finite set of possible values for the optical diameter is sorted out. However, such a feat would entail quite intensive and time-consuming computations, which are usually not at hand for in-flight data acquisition. Also, retaining the FWSCS response for all detected cloud particles proves impractical given the overwhelming file sizes that would be produced during a normal session of

measurements. For these reasons, it is common to discard the individual particle data once it is assigned to a size bin.

Nevertheless, certain CAS configurations allow for some sampling of the full particle-by-particle (PbP) data to be retained in dedicated output files. More precisely, the in-flight measurements are structured in finite time intervals called sampling instances, most conveniently one second long. In normal clouds, large numbers (frequently

several thousands) of droplets can be detected and measured during such sampling instances. The in-flight processing software may allow the storage of the FWSCS data for the first few hundreds (e.g. the first 292) of each sampling instance. A separate PbP output file containing this data is subsequently generated. As the selection of the particles contributing to the PbP data file appear to be completely random, their set can be considered as statistically representative for the entire set of detected particles during a measurement session. This assumption

is fundamental for our proposed use of PbP data in detailed post-flight analyses.

In the following sections we present a methodology for obtaining detailed droplet size distributions from such PbP sample files. This study relies on an analysis of a "most detailed" shape of the FWSCS-diameter diagram for pure water. As all local "ripples" of this diagram may play a role in sizing cloud droplets, it is concluded that the size distributions of various cloud parameters, as well as their bulk values, are most accurately expressed in post-

flight analyses by using "the finest" equidistant division (or mesh points) of the range of particle diameters. Nevertheless, for certain purposes, droplet distributions over coarser size grids (which are readily available from the "basic" ones obtained over the finest set of equidistant mesh points) may prove more practical. An obvious example is the design of unambiguous divisions of the whole range of diameters into uneven size bins for use in in-flight recordings. Also, coarser equidistant size grids can be very convenient in post-flight error evaluation of

resulting cloud parameters.

The proposed methodology is illustrated on short (few minutes) selections of data recorded during previous measurement campaigns of water clouds with an airborne CAS instrument. Error assessment is also performed in detail for the results obtained with the considered example data.

Additionally, some possible influences of atmospheric aerosols on the outcomes of the CAS measurements are

discussed, with emphasis on the possible alteration of the optical properties of cloud droplets by the dissolving or the inclusion (starting from the nucleation step) of sub-micron particles of hygroscopic/hydrophilic aerosol. The importance of the optical properties of measured particles has been long addressed in the literature (Liu et al., 1974; Johnson and Osborne, 2011; Rosenberg et al., 2012), insisting mainly on aerosol sizing and on instrument calibration issues. The wavelength of the light provided by a CAS light source is normally chosen in a range where

pure water has virtually no absorption. The whole sizing procedure assumes that any measured particle is a droplet


of pure water and the FWSCS-diameter diagram, which is the basic comparison tool, is constructed for the specific case of pure water. It follows that a significant increase in the absorption and/or refractivity properties of "contaminated" cloud droplets may induce drastic changes in their sizing from comparisons with the pure water diagram. These changes, if very numerous, may further degrade the objectivity of CAS measurements, the ensuing

size distributions of cloud droplets and the values of important bulk cloud properties. To improve the reliability of such results, some complementary measurements are suggested.

## 2. The detailed shape of the FWSCS-diameter diagram

According to Mie theory, the differential scattering cross section of light on dielectric spheres with given complex refractive indices is a complicated function of both the scattering angle and the diameters of the scatterers. The

details of this formalism can be found in any classic book on the subject (e.g. Bohren and Huffmann, 1983), so its derivation is omitted in this paper. As the intensity of the scattered light cannot be determined at a specific value of the scattering angle, any instrument used for such measurements is designed to capture the scattered light in a certain angular interval. The standard CAS collects the forward scattered light. Therefore, its sensors usually cover a small (around 10°) angle near the direction of the incident laser beam. It follows that the CAS is actually

measuring an integral of the differential scattering cross section over that specific angular interval. The value of this integral is what we call the FWSCS. We mention here in passing that the FWSCS still retains a quite strong sensitivity on the limits of the collecting angular interval, especially on its upper bound, so the accurate knowledge of these constructive parameters is of utmost importance for an objective use of the instrument. Our computations of the FWSCS have been performed through integration over the fixed angular interval stretching from 4.0° to

13.5°. It is also worth mentioning here that the wavelength used in FWSCS calculations was $\lambda = 658$ nm at which our instrument operates, according to its technical specifications. At this wavelength, pure water is almost non-absorptive (more precisely, the real and imaginary parts of the refractive index are $n = 1.331$ and $k = 2.23 \times 10^{-8}$, respectively).

Turning now to the FWSCS-diameter diagram, any type of CAS instrument determines the size of cloud particles

through comparison between measured and theoretical values of FWSCS, thus attempting an inversion of the FWSCS-diameter functional dependence. The characteristics of this dependence are therefore of paramount importance in the numerical phase of the CAS sizing process. Nevertheless, the theoretical FWSCS-diameter relationship is too complicated to be cast in a closed analytical form and it should be calculated point by point for a certain set of diameter values. As mentioned before, the typical range of diameters of particles detected by CAS

is 0.5-50 μm. The FWSCS diagram can be computed within this fixed range, at a certain number of equidistant mesh points, $N_d$. Constructing the FWSCS curve for increasing values of $N_d$ uncovers more and more "ripples" of it, as illustrated in Figure 1, which shows a close up of the dimensional range of scatterers between 37 and 40 μm. The figure's main panel contains the corresponding segment of the FWSCS-diameter curve plotted for three increasing values of $N_d$ (and thus for three increasing densities of mesh points on the abscissa). It can be seen that,

once $N_d$ increases, the curves look increasingly oscillatory at the local level. The presence of "ripples" clearly constitutes a difficulty in the process of retrieving a particle's size from a specific value of the FWSCS. For example, when assuming a measured value of $2.275 \times 10^{-6}$ cm$^2$ (indicated by the pink horizontal line in Figure 1), there are multiple possible values for the diameter of the scattering particle that produces such response. That





number is obtained by counting all of the intersections of the horizontal line in the FWSCS diagram. This number
obviously increases when the diagram is computed in greater detail. In older descriptions of the forward scattering
spectrometers (originally used for aerosol sizing measurements, see Baumgardner et al., 1992) this aspect seemed
to be overlooked and some smoothened versions of the FWSCS vs. diameter diagrams appeared to have been
used. More recent studies on OPCs (Rosenberg et al., 2012) consider in greater detail the consequences of the
non-monotonicity of the FWSCS vs. diameter correspondence, but focus mainly on the issues related to the
instrument calibration.

The obvious practical question arising in connection to the local irregularities of the FWSCS curve is: How fine
should the division of points on the abscissa be to reveal all the local features of its size dependence? In other
words, one should settle on a sufficiently large value of the number $N_d$ in order to have a reliable theoretical
FWSCS-diameter diagram that displays the full "noisiness" of this dependence. In order to answer this problem,
the diagram has been computed in the same range of diameters (0.5 to 50 μm), for an increasing sequence of mesh
point numbers. It was found that, with increasing $N_d$, the shapes of the resulted diagrams change, become more
detailed and increasingly similar. In quantitative terms, the similarity of two such diagrams can be measured by
the area enclosed between them. For convenience, these areas have been normalized to the area under the finest
plot in the sequence (namely that corresponding to $N_d$ = 10,000) and called "normalized similarities". Thus, for
each FWSCS plot in the sequence the "normalized similarity" with the preceding one in the sequence has been
computed. The "normalized similarities" have then been represented against their $N_d$ values and the result is
presented in the inset of Figure 1. It can be seen that the difference in shape between two successive FWSCS
diagrams drops close to zero for over 9,000 mesh points. As a consequence, a reference value of $N_d$ = 10,000
equidistant mesh points on the abscissa has been used in all following computations.

**3.        Retrieving particle diameters from the comparison with the FWSCS diagram**

As mentioned in the above discussions, the non-monotonicity of the FWSCS-diameter dependence induces an
important difficulty when extracting a particle's size from the FWSCS value it generates in a CAS instrument. To
be more precise, while there is an overall increase in the FWSCS values for larger scatterers, the dependence is
quite oscillatory and, at the local level, it shows a very noisy structure of small "ripples". Therefore, as shown
before in Figure 1, this makes the particle sizing highly ambiguous. A way to get around this difficulty would be
to use a coarser and uneven partition of size bins over the whole measurable range of particle diameters. As
mentioned before, the bin limits should be designed unambiguously, in the sense that the corresponding FWSCS
thresholds have unique intersections with the diagram. However, due to the high density of ripples in the FWSCS
curve, the possibilities of constructing strictly unequivocal (but still meaningful) divisions of bins are practically
quite limited. The difficulty may become more obvious when there is an interest in detailing regions of the cloud
droplets' dimensional spectrum, which fall in the noisiest parts of the FWSCS diagram. A straightforward
possibility to overcome this hurdle would be to use various smoothed versions of the FWSCS-diameter diagram.
Numerical smoothing of a data set can be achieved in several ways, but only the resulting shape is relevant. The
smoothing should be performed in a balanced degree, such that it does not alter the main features of the functional
dependence described by the diagram. An illustration of this requirement is presented in Figure 2.

Here we consider three versions of the FWSCS diagram and select the maximum number of bins for each of them.
The first diagram (Figure 2a) is the "raw" FWSCS curve (computed at 10,000 mesh points on the abscissa) and



the other two (Figures 2b,c) are smoothed versions of it constructed by the so-called median smoothing method which averages the FWSCS values at a certain (odd) number of consecutive points on the abscissa. The number

of points over which the average is performed is called the smoothing window. The maximal bin configuration has been established for each case according to two constraints. First, as already mentioned, it was required that the horizontal lines drawn for each FWSCS threshold intersect the diagram at a single point. The second requirement was that the width of a size bin was not irrelevantly small (for example, if the width of a size bin turned out to be lower than 5 % of the value assigned to its upper limit then the bin was merged with its next

adjacent neighbour). In this way, a one-to-one correspondence is established between the FWSCS thresholds and the limits of the size bins. Figure 2 illustrates how the degree of smoothing of the FWSCS diagram influences the bin width and spacing. When applying the bin construction method to the raw FWSCS diagram, a maximum of 13 bins is obtained (shown in Figure 2a and detailed in Figure 2d), which can be too coarse, especially for obtaining meaningful information about particles of larger sizes.

By applying a 3 points smoothing window, the number of bins increases to 17 (Figures 2b and 2e). Moreover, if the smoothing window is increased to 7 points, the number of bins becomes more representative across the entire measurable particle range (22 on Figure 2c) and more detailed information about the larger particles is retained. This improvement, however, is obviously obtained at the expense of distorting the local structure of the FWSCS-diameter diagram (Figure 2f). If the degree of smoothing is pushed further, even larger amplitude undulations of

the curve are wiped out and its deviation from the raw diagram is enhanced. Consequently, the new FWSCS thresholds are more likely to be ambiguous if referred to the raw curve and this will directly impact the accuracy of cloud droplet sizing.

The splitting of the range of cloud particle diameters into relatively large and uneven size bins is clearly very helpful in practical in situ measurements as it both allows for rapid in-flight counting and sizing of cloud particles

and saves storage memory by generating relatively small data files. The process of generating the size bin structure automatically produces the corresponding FWSCS grid of bins, which in principle allows for the rapid sizing of every valid particle that was sampled by the instrument, with no need to retain the particular value of the FWSCS it produced. The resulting statistics are usually expressed in various histograms (normalized or not) over the pre-established structure of size bins. Nevertheless, the practical procedure is not so simple as the FWSCS of a particle

is quantified in voltage counts of some specialized sensors (Baumgardner et al., 2001). Thus, the theoretical FWSCS grid of thresholds must be made to correspond to a grid of threshold counts of the sensor. This process involves precise knowledge of some electronic parameters of the instrument (since generating the voltage signals usually requires different non-linear amplification stages) and of certain specific constructive parameters of the instrument (like the angular collecting range of the scattered light, the effective sample area, the laser wavelength

etc.). Therefore, it is expected that constructing the sequence of threshold voltage counts to be associated with the size bin structure brings a certain amount of error that may be difficult to evaluate for older versions of such instruments (when some constructive parameters were quite poorly defined). Due to these difficulties, the rapid, in-flight statistical analysis of cloud particles should be complemented, whenever PbP FWSCS recordings are available, with post-flight PbP analysis.





**4.        Expressing cloud particle statistics from PbP files by using a fine grid of particle diameter values**

As discussed in the previous section, while rapid and memory saving, the use of uneven size bin structures constructed from the unequivocalness requirement inherently lose important details about the particle size distributions, especially at larger sizes (where the FWSCS diagram is more "rippled"). These ensuing errors are critical when size distributions are used to derive quantities that are highly sensitive to larger particles like the

liquid water content (LWC). To overcome such drawbacks, we propose the use of a dense mesh of a convenient number of equal size bins. This number of bins could be as large as that of the number of mesh points on the abscissa (10,000) for which the shape of the FWSCS-diameter diagram stabilizes, as discussed in Section 2. Further, consider a certain measured value of the FWSCS (i.e. a given entry of the PbP file). When theoretically computing the FWSCS, the same value is obtained for a number of values, $n_0$, of the particle diameter (they

correspond to all intersections of the measured value with the FWSCS diagram), each one of which is a possible optical diameter of the particle that produced the measured FWSCS response. As we lack information on which of these alternatives is more probable, we are forced to assume that any of them is equally possible. In other words, given the measured value of the FWSCS, we count $\dfrac{1}{n_0}$ for each diameter where the computed diagram takes on that value. In this way, we replace integer particle counts in unequivocal (but large and uneven) size bins with

fractional particle counts associated to each mesh point of the dense, equidistant grid of diameters. Detailed pointwise size distributions can thus be constructed. They can be used as they are but can be also grouped in any structure of bins, including that resulting from the smoothing of the FWSCS diagram, as described in the preceding section. The most important advantage of the proposed approach is that, instead of an uneven grid of size bins, one may use a structure of a convenient number of equal size bins (e.g. one micron wide). In most situations,

when presented in this way, the size distribution retains its information across all measurable sizes.

As already stated, the PbP output files contain detailed records on a certain subset of the entire group of measured particles in a given flight segment. The particles that enter the instrument during a flight segment and qualify for FWSCS measurement are normally so numerous that they can be treated as a statistical ensemble. The particles of which FWSCS values have been recorded in the PbP files are selected only on their arrival time in the sample

volume of the CAS. As for the selection of these particles no size criterion has been imposed (except for that of fitting into the 0.5 μm – 50 μm measuring range of the instrument), it follows that their sets will bear the same size statistical specifics as the entire ensemble. These size statistical peculiarities may include eventual dimensional gaps that can occur due to various effects, e.g. cloud mixing (Beals et al., 2015). This observation is key in acknowledging the usefulness of the PbP data. As noted in this section, detailed, pointwise size distributions

can be obtained from the PbP output files. By further grouping these distributions over the very (uneven) bin structure that served for the in-flight bulk data recording, one may compare the ensuing size distribution with that provided by the instrument (if both are properly normalized, for example at the total number of particles in each recording). If consistent, the PbP sample data should generate size distributions similar to those produced by the bulk data file. Such consistency checks of the PbP files can be easily performed for each sequence of data to be

processed through post-flight analysis in order to remove eventual accidental artefacts.

Moreover, if normalized to the so-called sample volume of each recording, further information can be very conveniently retrieved from the comparison of the aforementioned size distributions. The sample volume of some recording is the total volume of air that is transiting the instrument during that particular recording. For the bulk



data of a certain flight segment, the sample volume (to be denoted by $V_s^{tot}$) can be determined quite easily as the
product of the total duration of the flight line, the air speed and the physical area in which particles are detected
in the instrument (the sample area, as will be further discussed in Section 5). By contrast, the sample volume of
the set of particles that generate the PbP file (denoted simply by $V_s$) is more difficult to retrieve since the selection
of these particles is not continuous along the flight segment. By assuming that the total set of particles measured
in a flight segment and the selected subset that generates the PbP file have similar statistical behaviours, one may
compare their related size distributions over the uneven bin structure used for the in-flight bulk data recording.
Using the normalization of these distributions at the corresponding sample volumes, their shapes should be, in
principle, identical. At this point, one can use $V_s$ as an adjustable parameter and compute its value from the
condition that the "distance" between the two distributions (defined as the root mean square of their bin
differences) is at a minimum. The accuracy of this procedure can be further improved by using LWC size
distributions (also normalized at the corresponding sample volumes) instead of number size distributions. The
size distribution of the LWC over the uneven bin structure used in the rapid in-flight measurements is readily
obtainable from the corresponding number size distributions by multiplying it with the central water droplet mass
of each size bin (i.e. the mass of a spherical water droplet with the diameter equal to the median of the bin).

The aforementioned methodology has been applied on some data recorded by our group during recent
measurement campaigns. Our instrument, a CAS with depolarization (CAPS-DPOL) produced by Droplet
Measurement Technologies Inc. (DMT), in 2011, is mounted on a Beechcraft C90 GTx aircraft. The data files are
usually quite large, but the post-flight analysis has mainly focussed on selected segments where the aircraft flew
in warm clouds at approximately constant altitudes, thus probing various horizontal transects of the cloud. Such
segments are hereafter referred to as flight lines. To illustrate the typical post-flight analysis that we performed
with our recorded data, a single example will be discussed in this section. The data was collected in a flight line
during a measurement campaign performed in September and October, 2019, over Romania. A further example
is additionally used in the next section and refers to similar measurements performed in April, 2019. A
comprehensive description and discussion of these campaigns will follow in a forthcoming paper. The PbP file
contains detailed values of the measured FWSCS detector counts for 71,014 particles, which represent an excerpt
of the total number of particles that were validated and classified amongst a predefined size bin structure.
According to the DMT procedures, the result of this classification is recorded in a separate output file without
retaining any specifics for individual particles.

Applying the recipe described in this section to the PbP data recorded in the example flight line, the related detailed
size distributions are constructed over a typical fine grid of 10,000 values for the particle diameters, between 0.5
and 50 μm, which is the measuring range of our instrument. The significant parts of the number and LWC size
distributions are shown in the panels (a) and (c) of the Figure 3. The most striking aspect related to these diagrams
is the display of fine structures showing certain dimensional preferences (or "modes") of the cloud droplets.
Highlighting such peculiarities by in situ measurements might prove useful for correlating cloud microstructure
with the properties of the aerosol particles that are present in the studied area (as long suggested in the literature
– e.g. Squires, 1952; Mordy, 1959; Sorjamaa et al., 2004) as well as other cloud microphysical processes.

Such detailed analysis of in situ collected data could not be possible if the distributions were constructed over
coarser size bins. This point is illustrated in the panels (b) and (d) of Figure 3, where the same statistics as those
presented in panels (a) and (c) have been built over an equidistant grid of bins, each of almost 1 μm in length.





This grid is almost 200 times coarser than the detailed one, but still preserves some major features of the two
distributions. By contrast, the usual CAS acquisition software allows for only 30 (uneven) size bins where the
probed particles can be distributed. It is obviously expected that such coarse grids may further smoothen the
detailed shapes of the distributions and this aspect is made particularly clear by examining the plots of Figure 4.
Here, in panels (a) and (b), the detailed and the 50 equal size bins distributions have been reproduced from Figures
3a and 3b, respectively. For comparison, the plot of Figure 4c shows the size distribution from the same PbP file,
but represented over the 30 bins structure used for the in-flight data acquisition. Due to the small widths of the
bins in the sub-micrometer range, the distribution of Figure 4c has some resemblance to the detailed one of Figure
4a, in the same region. However, between 8 and 20 μm (a particularly important size range for the microphysics
of warm clouds), the 30 bins distribution looks rather "dull" and clearly lacks the structural richness of the
representation shown in Figure 4a. Nevertheless, as already pointed out before, coarser bin structures allow rapid
in-flight processing, use shorter data files and may serve for validating the PbP recordings as well as for computing
specific functional parameters.

As a conclusion, using the PbP data files, the methodology described in this section allows the construction of
detailed size distributions of cloud droplets if accurate descriptions are needed. Such procedure may be useful, for
example, in precise instrument calibrations (Rosenberg et al., 2012). Coarser size distributions (over equal or
uneven bin structures) are also readily available from the detailed one, for use in computing various cloud
microphysical parameters.

## 5.    General expression of cloud microphysical parameters and the ensuing errors

Mathematical expressions for microphysical quantities of clouds (like droplet effective diameter or LWC) usually
contain various averages over the size distributions of droplets. Each such average implies a summation over the
values taken by a certain function of droplets' diameters, $y(d)$. The sum (whose value we denote by $Y$) can be
generally written as:

$$Y = \int_{d_0}^{d_m} y(x) c(x) dx,$$

where $c(x)dx$ is the number of particles detected in the diameter interval $(x, x+dx)$ per unit of explored volume
(the so-called number concentration of particles). The integration is over the maximal interval $(d_0, d_m)$ within
which the particle diameters can take values (in our case, it is the measurement range of the instrument, namely
$d_0 = 0.5$ μm and $d_m = 50$ μm). For example, to obtain the LWC, the function $y(x)$ in Eq. (1) should be replaced by
some constant multiple of $x^3$. Also, for computing the extinction coefficient from its practical expression
(approximated as twice the optical cross section for cloud droplets and visible wavelengths – see, for example
Kokhanovsky, 2004), $y(x)$ in the integrand of Eq. (1) should be $x^2$. For computing the effective diameter ($d_{ef}$) of
droplets, one should consider a more complicated expression involving a ratio of two integrals of the type shown
in Eq. (1): one with $y(x) = x^3$ divided by the other with $y(x) = x^2$. Nevertheless, the following discussions essentially
apply for this case too.

Eq. (1) is written in the assumption that the number concentration is known for every value of the diameter.
However, as already discussed, all practical size distributions are discrete functions, over some finite number of
bins (evenly, or unevenly spaced), to be denoted by $N_b$. In passing, we may note that $N_b$ should not exceed the



total number of mesh points, $N_d$, on the abscissa where the detailed FWSCS diagram was computed. For such discrete distributions, the integral in Eq. (1) can be approximated by the corresponding sum over these bins:

$$Y = \sum_{i=1}^{N_b} y(d_i) c_i \,,$$

(2)

where $d_i$ and $c_i$ are the representative diameter (e.g. the median) of the size bin number $i$ and the number concentration of particles found in that bin, respectively. At this point, it is useful to detail explicitly $c_i$ by using the value of the sample volume, $V_s$, which can be obtained through the procedure described in the previous Section. Thus, we should write

$$c_i = \frac{N_i}{V_s} \,,$$

(3)

where $N_i$ is the total number of particles detected in the $i$-th bin (the sequence of all these numbers represents what is usually called the number distribution of particles over the given size bins). Therefore, Eq. (2) becomes:

$$Y = \frac{1}{V_s} \sum_{i=1}^{N_b} y(d_i) N_i \,.$$

(4)

Along with computations of bulk quantities of the type defined generically in Eq. (1), it is also necessary to evaluate the related error interval or, equivalently, the absolute error $\delta(Y)$. A natural and reliable approach would be to first compute the relative error of the quantity $Y$:

$$\varepsilon(Y) = \frac{\delta(Y)}{|Y|} \,.$$

(5)

As the error analysis is simpler in continuous variables, we return to Eq. (1), which can be more conveniently detailed in the following form:

$$Y = \frac{1}{V_s} \int_{d_0}^{d_m} y(x) N(x) dx \,,$$

(6)

where $N(x)dx$ is the number of detected particles having the diameters in the range $(x, x+dx)$. The error in $Y$ originates partly in the imprecision of determining the sample volume $V_s$. The other source of $\varepsilon(Y)$ originates in the error of each measured value of the FWSCS, which translates in a complex way to the number distribution $N(x)$. When assuming such an imprecision for the FWSCS optical measurements, the distribution $N(x)$ takes another shape and shifts to a new function denoted by $\tilde{N}(x)$. The shift between these two distributions should have no constant sign over the whole range of diameters. On the contrary, their difference should oscillate around zero as any overestimation of the particle number in a certain size range should induce an underestimation somewhere else. We can therefore write:

$$\varepsilon(Y) = \varepsilon(V_s) + \frac{\delta\left( \int_{d_0}^{d_m} y(x) N(x) dx \right)}{\left| \int_{d_0}^{d_m} y(x) N(x) dx \right|} \,.$$

(7)

Moreover,





$$\delta\left(\int_{d_0}^{d_m} y(x)N(x)dx\right)=\left|\int_{d_0}^{d_m} y(x)\left[\tilde{N}(x)-N(x)\right]dx\right|. \tag{8}$$

Thus,

$$\varepsilon(Y)=\varepsilon(V_s)+\frac{\left|\int_{d_0}^{d_m} y(x)\left[\tilde{N}(x)-N(x)\right]dx\right|}{\int_{d_0}^{d_m} y(x)N(x)dx}. \tag{9}$$

As already discussed, integrals like those appearing in Eq. (9) can be practically computed by summing over some custom grid of size bins. If the grid were made of uneven bins, then the errors ensuing from the eventual smoothing procedure of the FWSCS diagram (which, in some cases, could be quite consistent) should also be taken into

account. To avoid such artificial extension of the overall imprecision, a grid of equal bins (which is defined, so it is not affected by errors) is normally recommended. Using such discretization of the range of diameters in equal bins, one obtains:

$$\varepsilon(Y)=\varepsilon(V_s)+\frac{\left|\sum_{i=1}^{N_b} y(d_i)\left(\tilde{N}_i-N_i\right)\right|}{\sum_{i=1}^{N_b} y(d_i)N_i}. \tag{10}$$

For the case of $d_{ef}$ which can be computed through the relation:

$$d_{ef}=\frac{\sum_{i=1}^{N_b} N_i d_i^3}{\sum_{i=1}^{N_b} N_i d_i^2}, \tag{11}$$

the relative error takes a form that is independent of the imprecision in $V_s$:

$$\varepsilon(d_{ef})=\frac{\left|\sum_{i=1}^{N_b}\left(\tilde{N}_i-N_i\right)d_i^3\right|}{\sum_{i=1}^{N_b} N_i d_i^3}+\frac{\left|\sum_{i=1}^{N_b}\left(\tilde{N}_i-N_i\right)d_i^2\right|}{\sum_{i=1}^{N_b} N_i d_i^2}. \tag{12}$$

where $\tilde{N}_i$ is the "distorted" distribution of particles over bins.

In this way, the remaining problem is to obtain $\tilde{N}_i$ as generated by the error associated to each experimental

FWSCS. To make any meaningful progress in this difficult matter, one should actually resort to evaluating the related maximal distortion error. Such an attempt could be imagined as follows: For any measured value, $C$, of the FWSCS, there should be an assumed absolute error, $\delta C$. We will also assume that the true value can be found, with uniform probability, somewhere in the horizontal strip defined by the interval $\left[C-\frac{1}{2}\delta C,\ C+\frac{1}{2}\delta C\right]$ in Figure 5. In other words, instead of obtaining a sharp, exact value $C$ for the FWSCS, the instrument provides a

"blurred" figure of width $\delta C$. It is obvious that, the wider the error strip for $C$ is, the larger the imprecision of the



particle sizing will be. For this reason, it is further assumed that the maximal distortion of the size distribution from the one obtained with "exact" values of the FWSCS results by counting the "blurred" intersections of the FWSCS-diameter diagram with the error strips associated with each measured particle. To proceed in this computation, consider first the intersection of the horizontal line at $C - \frac{1}{2}\delta C$ with the FWSCS diagram. Let the

abscissa of that point be denoted by $d_{min}$. Also, name $d_{max}$ the abscissa of the rightmost intersection of the horizontal line at $C + \frac{1}{2}\delta C$. As clearly illustrated in Figure 5, not all the points of the FWSCS diagram with abscissae between $d_{min}$ and $d_{max}$ fall within the error strip (for example, points with $d$ around 40 μm are not included). Now, imagine that we remove from the interval [$d_{min}$, $d_{max}$] all the abscissae for which the FWSCS values fall outside the error strip. The remaining set, which is actually a union of smaller intervals, will be called $\Delta$. As the true value

of the measured FWSCS is assumed to be somewhere in the (yellow) strip defined by the interval $\left[ C - \frac{1}{2}\delta C, \, C + \frac{1}{2}\delta C \right]$, it is clear that the true value of the particle's diameter associated with the value $C$ of the FWSCS should lie within the set $\Delta$. According to our assumption, every value of the FWSCS within the error interval $\left[ C - \frac{1}{2}\delta C, \, C + \frac{1}{2}\delta C \right]$ has the same chance of being the true one. On the other hand, every horizontal line drawn within the error strip will intersect the FWSCS in a unique set of points, with a unique set of abscissae.

However, from one horizontal line to another, the number of intersections may differ (depending on the local shape of the FWSCS diagram), so there should be different chances that one point or another from the portion of the FWSCS within the error strip corresponds to the true diameter. The same should be valid for the corresponding weights with which the particles' diameters enter in the counting of each size bin. To quantify the weight for a given size bin, one might select from the set of all intersection points of the FWSCS diagram with the error strip

only the set of points whose abscissae fall inside that size bin. Then, the required weight will be the ratio of the measures of these two sets of points of the FWSCS diagram, namely the smaller one to the larger one. Unfortunately, the usual representations of the FWSCS curves are not metric spaces, so one cannot simply use the length of the curve as a measure of a set of its points. Instead, one could rely on the "ordinate length" of a certain segment of the curve. This quantity can be defined as the sum of the absolute values of the ordinate projections

of all monotonic parts of the curve within that segment. Thus, we can denote by $\Phi_{err}$, the ordinate length of the portion from the FWSCS diagram that fit within the error strip and by $\Phi^i_{err}$, the ordinate length of the subset of the diagram that fit within the error strip and have the abscissa projections within the size bin number $i$. The desired weight with which the given particle contributes to that size bin can then be defined as the ratio $\Phi^i_{err}/\Phi_{err}$. Moreover, by summing up these weights for all measured particles, one can obtain the "distorted"

number of droplets with sizes contained in the bin number $i$, $\tilde{N}_i$. As the distribution $\tilde{N}_i$, obtained in this way, accounts for the maximal imprecision of each FWSCS measurement, we assert that it represents the maximal departure from the "correct" distribution $N_i$. If replaced in Eq. (10), it will provide the maximal relative error of the quantity $Y$. Increasing the upper bound of the error by using the inequality





$$\sum_{i=1}^{N_b} y(d_i)\left|\tilde{N}_i - N_i\right| \geq \left|\sum_{i=1}^{N_b} y(d_i)(\tilde{N}_i - N_i)\right|$$ at the numerator of the second term of Eq. (10) would mean

accepting the exceptional possibility that the terms of the sum $\sum_{i=1}^{N_b} y(d_i)(\tilde{N}_i - N_i)$ are all positive. However,

in the case of number distributions, this situation can never happen due to the condition that the total number of

particles is the same, irrespective of the way they are distributed over the bins: $\sum_{i=1}^{N_b} \tilde{N}_i = \sum_{i=1}^{N_b} N_i$ . Thus, some

terms are necessarily negative and therefore, while the inequality $\sum_{i=1}^{N_b} y(d_i)\left|\tilde{N}_i - N_i\right| \geq \left|\sum_{i=1}^{N_b} y(d_i)(\tilde{N}_i - N_i)\right|$

is formally correct, its left term would lead to a physically overrated upper bound of the error.

By attempting to apply the above recipe for error evaluation to a too detailed distribution (as are the examples
shown in panels (a) and (c) of Figure 3, one may readily conclude that the computational effort is inconveniently
large, as usually the analysis extends over multiple flight lines.

As pointed out in this section, one of the most difficult task in error evaluation is computing the maximally
distorted size distribution. On the example flight line used in Section 4, the distortion has been computed for the

distribution over the equidistant structure of 50 bins. The result is plotted as a histogram in Figure 6, together with
the "exact" (or "nominal") distribution (the same as that appearing in Figure 3b) for comparison. One should note
that the differences between the two distributions may be locally quite large, although the logarithmic scale of the
ordinate might diminish their appearance. The distorted distribution has been evaluated with the hypothesis that
FWSCS measurements bear a homogeneous overall error of 10 % from the nominal values, which, for our

instrument, is well below the manufacturer's estimations. Nevertheless, the errors in FWSCS measurements (more
precisely in the numbers of "counts" given by instrument's detectors at each measurement) may actually depend
on various conditions (e.g. on the gain stages used in a given measurement) and cannot be taken as fixed at, say,
10 %. To evaluate the impact of the accuracy in FWSCS measurements over the bulk parameters of the clouds,
we computed the ensuing relative errors induced in three such quantities (namely the LWC, the extinction

coefficient and the effective diameter) for a range of values of the relative errors in measuring the FWSCS. It can
be seen in Figure 7 that, as expected, the increase in the imprecision for FWSCS make the errors of all bulk cloud
parameters grow. According to the above discussions in this section, to compute relative errors of LWC and
extinction coefficient one needs to evaluate the relative error in the value of the sample volume for the particles
involved in the PbP recording, $V_s$. Therefore, there will always be a background error for such bulk parameters.

In Section 4 we described a simple and reliable procedure of obtaining $V_s$ by comparing PbP vs. bulk data size
distributions over the operational in-flight structure of size bins. From this method, $V_s$ results as a certain fraction
of $V_s^{tot}$. Therefore, the relative error of $V_s$ should be the sum of the relative errors of $V_s^{tot}$ and of the fraction itself.
The fraction error is essentially stemming from a comparison between the two size distributions and it will be
assumed negligible. Consequently, the relative error of $V_s$ will be taken as that of the sample volume of the whole

recording in the given flight line, $V_s^{tot}$. This parameter is actually a composite one, as it requires the knowledge of
the velocity of the airflow in the instrument (the so-called probe air speed, or PAS), the duration of the
measurements and the so-called sample area, which is the physical area where particles are detected. This last
quantity should be, usually, provided by the manufacturer. The output files constructed by the processing software





of CAS-DPOL typically provide the PAS at fixed time intervals (sampling instances, e.g. one second each). Moreover, another string of the bulk data file generated in-flight from all validated particles provides the final moments of each sampling instance and is called "End Seconds". These entries can be used to extract the exact durations of the sampling instances for the whole bulk recording. By multiplying these time intervals with the corresponding PAS values and with the assumed value of the sampling area, one readily obtains a string of sampling volumes to be associated with the corresponding sampling instances and, by summing them up, the

flight line sample volume, $V_s^{tot}$, is obtained. Due to the large imprecision in the knowledge of the sample area (Lance et al., 2010), the relative error of $V_s^{tot}$ has been settled at 20 % in all cases considered in this study.

It should be noted here that, in principle, $V_s$ could result from a string of the PbP output file which records the time separations between successive particle measurements (the so called "inter-arrival particle time", or IPT). However, as already mentioned, the PbP data is recorded only for the first ~ 290 particles detected in a sampling

instance and the IPT is retained for each measured particle, without counting the "jumps" between successive sampling instances. This particular feature actually hinders the use of the IPT data string for reliably computing $V_s$ and underscores the utility and simplicity of the method described in Section 4.

Overall, we can conclude that evaluating the accuracy of cloud microphysical parameters obtained from CAS measurements is no straightforward matter. It involves a complicated and time consuming analysis of the PbP

files and relies on the knowledge of the detecting precision of CAS for individual particles, as well as on the precise knowledge of constructive parameters of the instrument (e.g. the effective sample area).

### 6.      The effects of increased droplets' refractivity and/or absorption on their sizing

As discussed in the previous sections, due to the complicated shape of the FWSCS vs. diameter diagram, which is at the core of the numerical phase of the CAS method, the whole procedure of sizing cloud particles is far from

straightforward. Additional uncertainties may also stem from the assumption that the measured particles are pure water droplets.

In fact, real cloud droplets are necessarily "contaminated" by aerosol particles, either by incorporating or by dissolving them (or even both) and it might be suspected that the forward scattered light from such complex particles might differ from the case of pure water droplets of the same size. A convenient approach to describe

the optical response of "contaminated" particles is by using a composite refractive index, which is generally larger (in both its real and imaginary parts) than the one of pure water (Erlick, 2006; Liu and Daum, 2002; Wang and Sum, 2012; Mishchenko et al., 2014).

Moreover, the FWSCS-diameter diagram is quite sensitive to the values of both the real and imaginary parts of the particle's refractive index (Figure 8). Therefore, it may turn out that the sizing and classification of cloud

droplets may be flawed by using the pure water version of this curve. Indeed, as seen in Figure 8a, refractivity larger than that of pure water keeps the FWSCS-diameter diagram highly oscillating, but it gets an overall decrease with respect to that for pure water. The decrease is more significant for large droplets. On the other hand, changes in the absorption lead to more abrupt deformations of the diagram (Figure 8b). Even slight increases of the imaginary part of the refractive index produce strong distortions of the curve in comparison to that for pure water.

There is also an overall decrease (which tends to be very large) and a smoothing effect at higher absorptivity and for large droplets. Based on such examples, one may conclude that, if regarded as made of pure water (as the CAS method does), "contaminated" cloud droplets appear generally smaller than they really are.



One very important observation is that even large increases of both refractivity and absorptivity have little effect on the sub-micrometer range of the FWSCS-diameter diagram. The small sensitivity of the FWSCS on the
absorptivity extends actually towards the 3 μm range. To better highlight this remark, the small diameter parts of the curves shown in Figure 8 have been zoomed in on Figure 9, where logarithmic scales have been used on the ordinates. It can be seen that, when the refractivity varies quite widely (Figure 9a), shifts no larger than 100 nm may occur in evaluating the sizes of sub-micrometer objects. Also, for large variations of the absorptivity (Figure 9b), the FWSCS diagrams almost coincide in most of the sub-3 μm range. This peculiar aspect has little
importance on classifying cloud droplets, since they rather seldom fall into that size range. Instead, the observation refers mainly to eventual aerosol particles that can be detected by CAS if they are larger than 0.5 μm. It means that fine, sub-micrometric, aerosol particles are most likely to be correctly sized by CAS, irrespective of their refractive indexes, as they are "seen" as a lump portion of the size distribution.

Once the effect of refractive index variations on the FWSCS-diameter curve is established, one may check
for the related distortions of the size distributions. As concluded after discussing the examples plotted in Figure 7, moderate increase in either the real or the imaginary parts of the refractive index of a cloud droplet makes it appear smaller in a CAS measurement. The overall consequence of this fact is that the size distributions of clouds with "contaminated" droplets as resulted from CAS data are somehow shifted towards smaller diameters and the size-related properties (like the LWC) are underestimated. To check the amplitude of such eventual distortions,
we considered the same PbP data file as that used for constructing Figure 3a and we analysed it with FWSCS diagrams computed with modified refractive indexes. The results are shown in Figure 10. Both detailed number (panels a-c) and LWC (panels d-f) size distributions have been computed for the cases when droplets would have had higher refractivity (Figure 10, b and e) or higher absorption (Figure 10, c and f) with respect to the pure water (Figure 10, a and d). In both cases, the distributions are right-shifted with respect to those obtained for pure water.
The shift is more obvious in a bulk size parameter like the effective diameter which should be larger with between 10 and 15 % if "contamination" of droplets is taken into account. In other words, this means that "contaminated" cloud droplets are systematically placed in a lower range of diameters when the CAS data is processed with the FWSCS diagram for pure water. The effect is more visible in the shift of the total LWC, as indicated in the panels d-e of Figure 10. It can be seen that rather realistic changes of either real or imaginary parts of the refractive index
(Erlick, 2006) may lead to underestimations of between 25 and 30 % of the total LWC when assuming that "contaminated" droplets have the optical parameters of pure water.

To better quantify the influence of the refractive and absorbing optical properties of the cloud droplets on their sizing by CAS and on the ensuing cloud parameters, the same PbP data file from September 26[th] has been processed using FWSCS diagrams computed with a relatively wide range of values for the real and imaginary
parts of the refractive index. Then, using the detailed size distributions obtained for each case, some of their bulk characteristics have been computed and plotted as three-dimensional diagrams against the real and imaginary parts of the refractive index. Figure 11 shows four such diagrams, namely for the $d_{ef}$ (panel a), for LWC (panel b – note here that we use g/m$^3$ as units since we refer to the total LWC), for the mean diameter (panel c) and for the extinction coefficient (panel d). The mean diameter is essentially defined as the arithmetic mean of the diameters
of all the droplets considered in the PbP file. As all these parameters are more sensitive to the imaginary part of the refractive index, $k$, this variable was shown in logarithmic scale. One can readily note steep jumps in all diagrams for $k$ beyond approximately the value of 0.001. Further increase in absorptivity above this threshold



becomes almost irrelevant for any of the computed parameters. Some of them, like the effective and mean diameter, only approximately double their values in their jumps. However, the quantities involving moments of the second and third degrees of the size distributions (like the extinction coefficient and the LWC) have much higher increases (about four and six times, respectively). The variations in the refractivity values (the real part of the refractive index) produces slighter increases in all the considered parameters. The results illustrated in Figure 11 show quite clearly that the CAS analysis of cloud droplets may become abruptly flawed when the particles are "contaminated" with light absorbents over some specific threshold. The droplets' sizes and, most importantly, the cloud parameters deriving therefrom, become largely underestimated.

The obvious question stemming from such results is on the degree of reliability of CAS measurements. First, it should be remembered that cloud droplets have been considered *all* "contaminated" to the same degree in our computations. This is obviously far from true. The aerosol incorporation should itself follow a certain distribution, dictated primarily by the variations of aerosol concentration along the cloud's vertical dimension. Moreover, in order to be incorporated in water droplets, the aerosols should be hygroscopic/hydrophilic, which is not always the case for strong absorbers in the visible spectrum. Also, to achieve absorptivity levels that make $k = 0.001$, the droplets should "ingest" light absorbing aerosols at relatively large concentrations (Erlick, 2006), which can be the case only in events of intense atmospheric pollution. Even in such situations, the volume fraction of "ingested" aerosol particles in water droplets is far from uniform. In a coarse estimation, this volume fraction should be proportional with $d^{-3}$, but also with $d^2$ (since the probability of each "ingestion" act should be proportional to the droplet's cross sectional area). Thus, unless the droplet results from coalescence of smaller ones that are already "contaminated", it may be concluded that the volume fraction of "ingested" aerosol should be roughly proportional to the inverse of the droplet's diameter. Implicitly, it follows that the impact of droplet "contamination" should be less severe on larger droplets. Such observations may obviously reassure CAS users that their measurements are generally quite objective. Nevertheless, to improve the reliability of the post-flight analyses, one should find useful to make assessments into the overall aerosol loading and composition. This task can be performed with complementary airborne instrumentation. Nevertheless, some primitive information can be also obtained directly from the PbP files generated by CAS. More precisely, estimations of the fine aerosol loading (the one that is more likely to be incorporated in water droplets) can result from closely inspecting the sub-micrometer tail (usually between 0.5 and 1 µm) of the detailed size distribution obtained from the PbP file. According to our previous remark (see the discussion related to Figure 9), the shape and position of this tail should not be affected much by the differences in refractive index between the aerosol particles and the pure water standard used by CAS. Therefore, the amplitude of the sub-micrometer (even sub-3 µm) part of the detailed distribution should, at least qualitatively, indicate the amount of the cloud's optical "contamination" and the level of confidence in the parameters computed from its detailed droplet size distribution obtained with the pure water FWSCS standard diagram.

Due to specific circumstances, the aerosol loading may differ at various altitudes in the cloud and this is a further reason for splitting a large PbP file into smaller parts corresponding to separate flight lines. An example is shown in Figure 12 for which the data has been collected during a flight on April 15th, 2019. Panels a, b, on one side and c, d, on the other side, show the detailed size distributions obtained from two flight lines, at different altitudes, through the same warm cloud. By comparing Figures 12b and 12d, it can be seen that, due probably to some aerosol advection, the sub-micrometer tail is significantly larger in the higher flight line than the one obtained



about 100 m below. One may therefore conclude that the CAS sizing in the upper line is less reliable than the one in the lower line. Nevertheless, such assertions should be enforced by more targeted complementary in situ measurements on the aerosol loading and composition in each circumstance. As instrumentation for such measurements is expensive and complicated to mount on small research aircraft, one could also rely on simpler, less precise methods. For example, it may be useful only to collect separate samples of liquid water from each flight line and to simply measure their refractive and absorptive properties for the same wavelength as that of the CAS's laser. These quantities are actually the only ones that are of importance for CAS sizing: they can be readily

used as average values for $n$ and $k$ in computing a more realistic FWSCS diagram used for evaluating each droplet.

### 7.     Conclusions

The present study details a numerical methodology for obtaining droplet size distributions from PbP sample files recorded with an airborne CAS instrument. First, we show how refined the size resolution should be in order to achieve a FWSCS diagram with reliable shape. The next step was establishing a procedure for obtaining particle

diameters from the measured value of the FWSCS, by comparing it with the one computed using Mie theory and the hypothesis that the instrument's laser beam scatters on pure water droplets. Cloud particle statistics could be further constructed using a fine grid on the diameter scale in order to capture the details of the size distributions. The possibility and the utility of coarser size grids, with either uneven or with equal bins was also briefly discussed. A small amount of PbP data obtained from our recent measurement campaigns has been used to

illustrate the proposed methodology. The general expressions of cloud microphysical parameters were written for size distributions over smaller numbers of equal bins and the ensuing relative errors were discussed in detail. It was thus shown that the overall uncertainties in computing various cloud parameters are mainly driven by the errors in measured FWSCS values for each particle. The influence of the relatively large imprecision in the values of the real and imaginary parts of the refractive index of cloud droplets on their size distributions and on the

ensuing cloud parameters was analysed in the final part. It was concluded that, when high atmospheric loads of hydrophilic and light absorbing aerosols are present, such imprecisions may drastically impact the reliability of the cloud data obtained through CAS measurements. Some possible complementary measures for improving the quality of the information obtained in post-flight analyses were suggested.

**Author contribution:** SNV and DEM formulated the project ideas, planned and performed the in-flight measurement campaigns for data acquisition. VF performed the data processing and developed the related algorithms. Further, VF prepared the first draft of the manuscript with support from all co-authors. ROD, TS, GI, AC and VV contributed to the interpretation of most results and, through numerous new ideas and observations, were instrumental in completing several successive versions of the manuscript. All the authors discussed the

results and contributed to the final manuscript.

**Competing interests:** The authors declare that they have no conflict of interest.



**Acknowledgements:** The authors acknowledge support of their work from the INCAS Nucleus project PN 19 01
08 03, part of the AEROEXPERT 2019-2022 program and from the project EEA-RO-NO-2019-0423/IceSafari,
contract no. 31/2020, under the NO Grants 2014-2021.

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



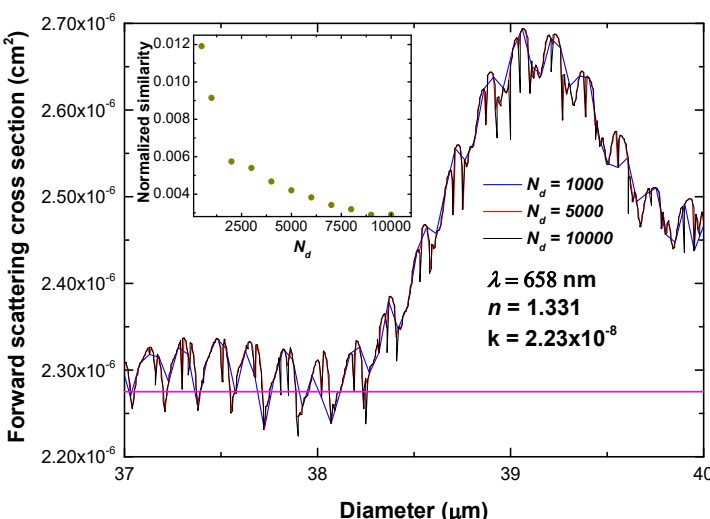

Figure 1: Detail of the FWSCS vs. particle diameter diagram plotted for three different values of the mesh point number, $N_d$, on the abscissa (the blue, red and black lines on the main panel). The wavelength of the radiation, $\lambda$, as well as the corresponding real ($n$) and imaginary ($k$) parts of the refractive index of pure water are indicated in the main panel. Increasing densities of these mesh points reveals more "rippling" in the structure of the curve. This, in turn, makes the particle sizing increasingly ambiguous. The pink horizontal line corresponds to a single measured value of $2.275 \times 10^{-6}$ cm$^2$ which intersects the FWSCS diagram multiple times. Thus, assigning a unique value for the diameter is impossible. The inset illustrates the analysis of the shape convergence of the FWSCS plots for increasing mesh points density on the abscissa.





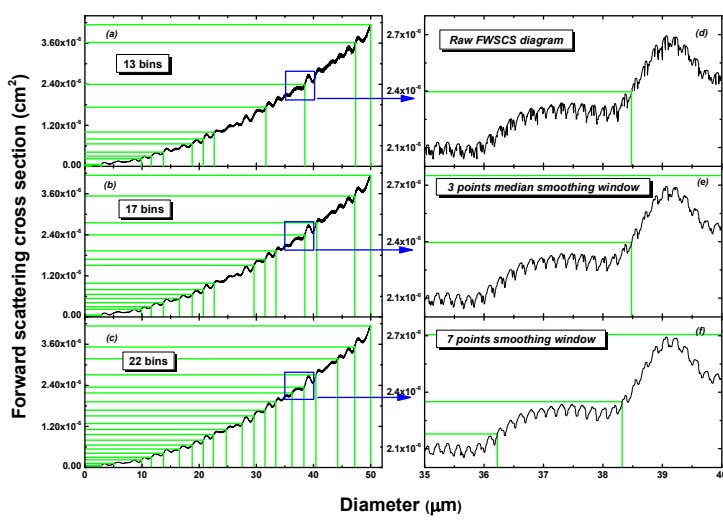

**Figure 2: Size and FWSCS bin structures induced on the FWSCS-diameter diagram. Green lines in panels (a), (b) and (c) indicate the size bins (on the abscissa) and the FWSCS bins (on the ordinate), for three degrees of smoothness, as indicated in panels (d), (e) and (f), where one same detail of the diagram is magnified for clarity.**

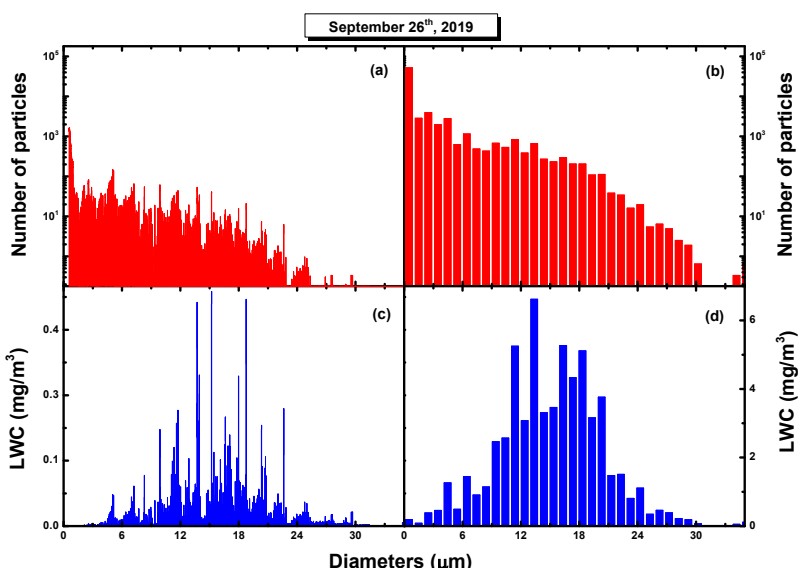

**Figure 3: Number (a, b) and LWC (c, d) size distributions obtained from the post-flight analysis of PbP data recorded during a flight line performed on September 26th, 2019, over some area of Romania. The diagrams (a) and (c) are detailed distributions and show a fine structure of size "modes", which are "wiped" out if spread over coarser size grids, as seen on the histograms (b) and (d) constructed over a structure of 50 equal size bins. Nevertheless, these coarser representations allow for sufficient resolution to accurately compute various averages and are more convenient in evaluating maximal relative errors, as described in the next section. Note the LWC unusual range of values due to the small amounts of liquid water counted in each division of the very fine grid of diameter values. Also, in panel (d), note that the levels of the LWC distribution are higher than those of the panel (c) as they collect the contributions of particles from larger divisions of diameter values.**



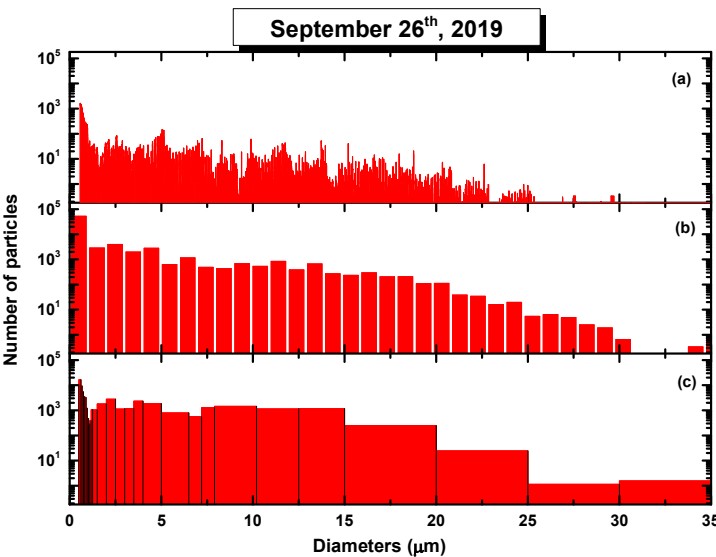

**Figure 4: Droplet size distribution of the Figures 3a and 3b (panels a and b, respectively) shown in comparison with the size distribution of the same droplets constructed over the 30 bins structure used for the in-flight data acquisition.**





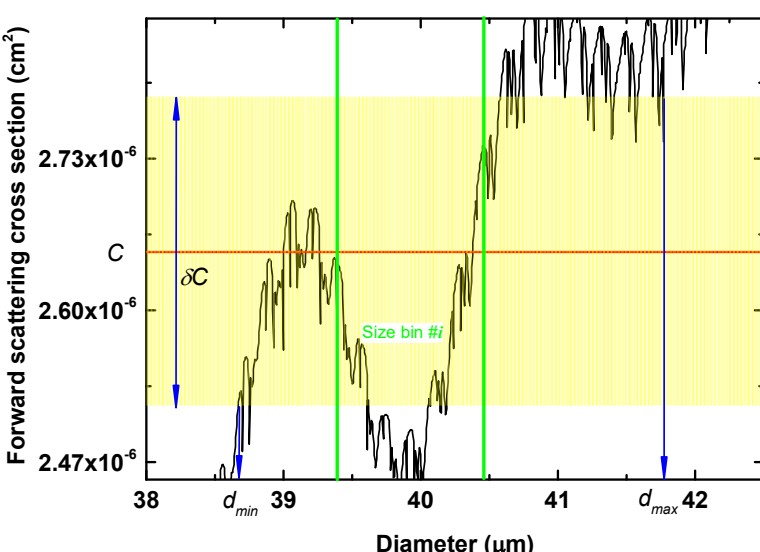

**Figure 5: Detail of the FWSCS-diameter diagram showing a measured value, *C* (red line), of a particle's FWSCS and the ensuing error strip (yellow region). The particle's contribution to the "distorted" size distribution can be computed by considering the "length" of the "blurred" intersection of the error strip with the FWSCS diagram. That "length", $\Phi_{err}$, is defined as the absolute values of the sum of the ordinate projections of all the monotonic parts of the diagram that fit within the error strip. If we further restrict to the part of the curve that fits within the error strip *and* in the size bin #*i*, then the analogous "length" $\Phi^i_{err}$ results. Its ratio to $\Phi_{err}$, is the weight with which the given particle contributes to the size bin #*i* (enclosed by the vertical green lines).**



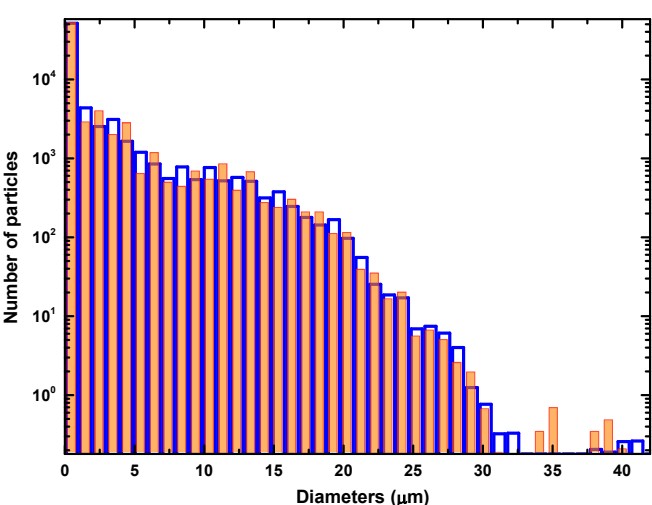

**Figure 6: Nominal (orange) and distorted (hollow blue) number distributions over a structure of 50 equal size bins. The distorted histogram is obtained with the assumption that FWSCS measurements have a homogeneous overall error of 10 % from the nominal values.**

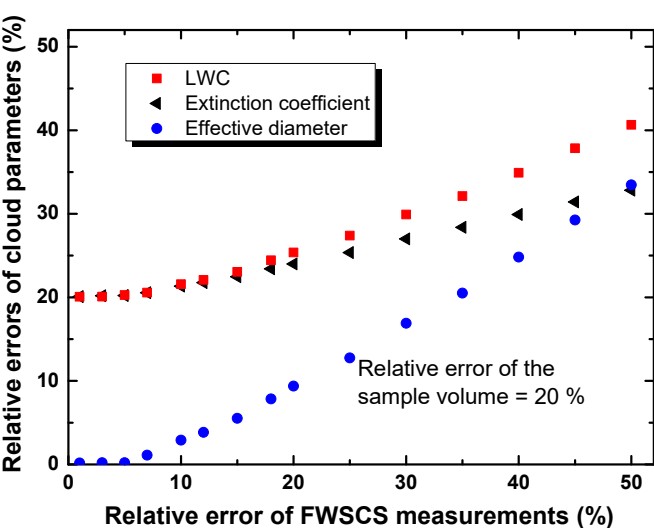

**Figure 7: Relative errors of the LWC, the extinction coefficient and the effective diameter as functions of the relative error in measuring the FWSCS. Different scales have been used on the axes in order to reach a convenient aspect ratio of the figure.**

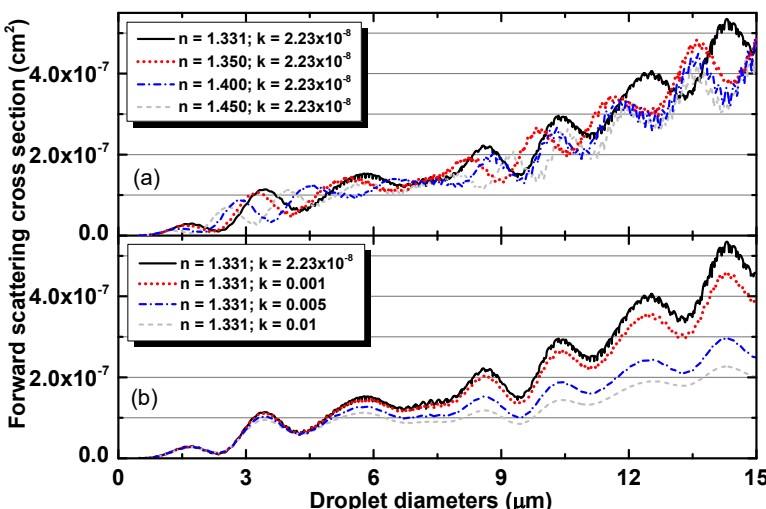

**Figure 8:** Computed FWSCS-diameter diagrams for the same size range, laser wavelength and angular interval of collecting scattered light, but for (a) different values of the real refractive index and (b) various degrees of absorption (as indicated by the imaginary part of the refractive index)





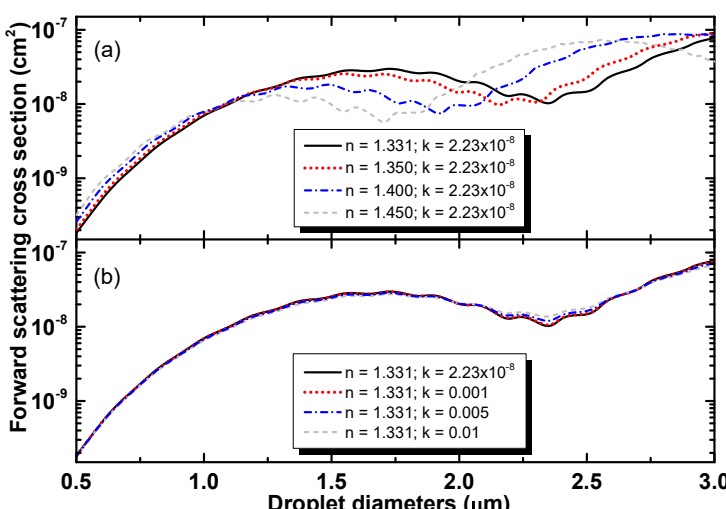

**Figure 9: Enlarged view of the small-diameter range of the diagrams of Figure 8, with focus on the sub-3 μm region. (a) Horizontal shifts no larger than 100 nm may occur when the real part of the refractive index varies. (b) Wide variations in absorption (imaginary part of the refractive index) make almost no changes in the zoomed region of the FWSCS diagrams.**

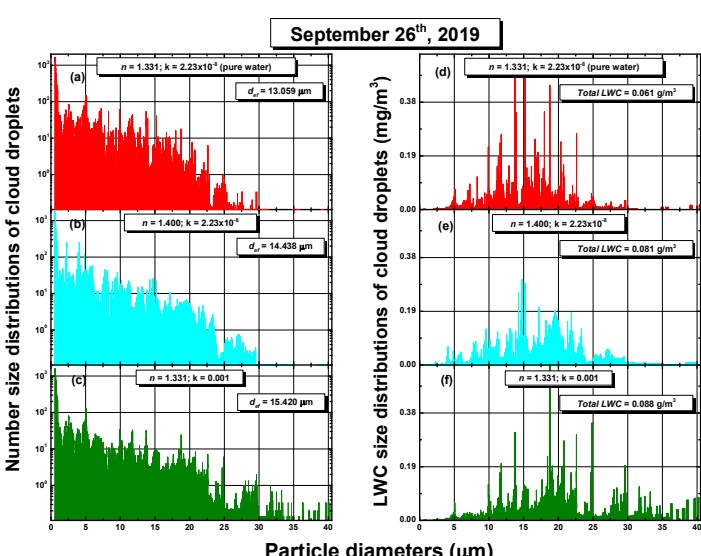

**Figure 10: Effect of eventual variations in the refractive index on the detailed number (a-c) and LWC (d-f) size distributions illustrated on data collected in a flight line performed on September 26th, 2019, over some part of Romania. Panels (a) and (d) are constructed using the refractive index of pure water. Both distributions shift to the right when either the refractivity (b, e) or the absorption (c, f) increase, indicating that using optical parameters of pure water leads to underestimations in both size and LWC. While the effective diameter, indicated in panels (a)-(c), is underestimated with 10 to 15 %, the underestimation of the total LWC, indicated in panels (d)-(f), runs between 25 and 30 % when assuming optical parameters of pure water for "contaminated" droplets.**



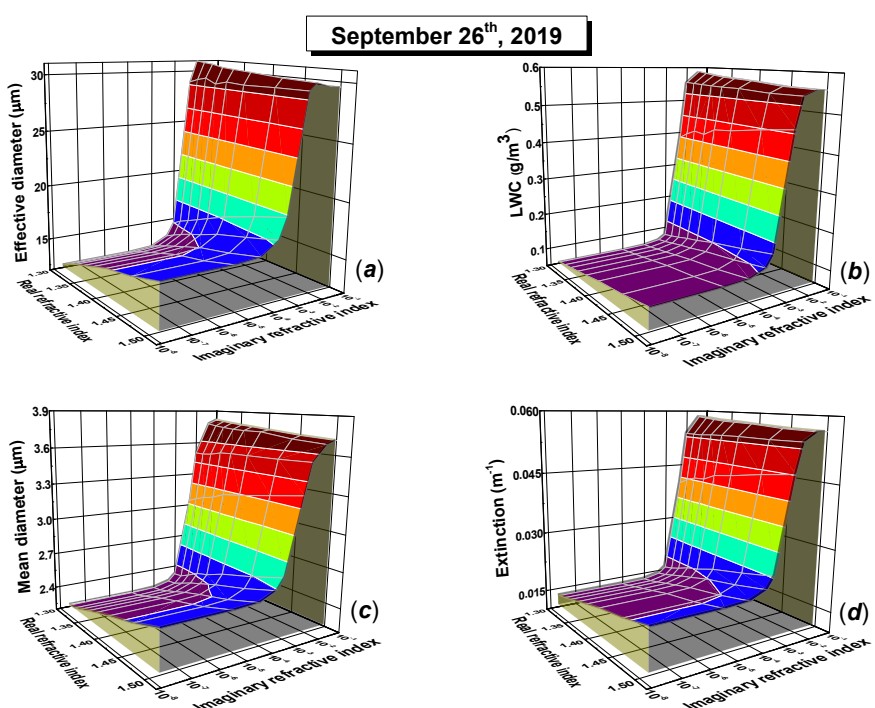

Figure 11: Effective diameter (a), total LWC (b), mean diameter (c) and extinction coefficient computed with the data used to construct Figure 3b for which the droplet sizing has been performed by assuming the particles have various values of the real and imaginary parts of the refractive index. Each diagram shows a steep increase for imaginary (absorption) index larger than about 0.001, which suggests that droplet sizing is largely underestimated in clouds "contaminated" with substances that highly absorb the wavelength of the CAS laser.



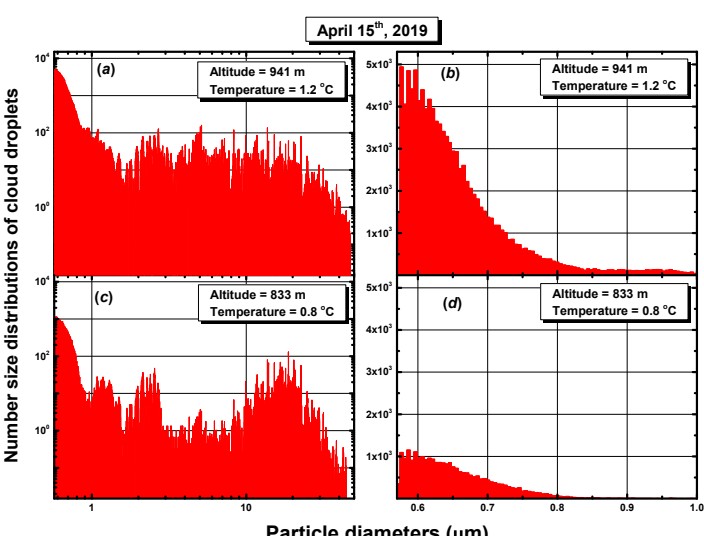

**Figure 12: Detailed size distributions obtained with the standard pure water FWSCS from two PbP files of different flight lines performed on April 15th, 2109, through the same liquid cloud over southern Romania. Panels (a) and (b) correspond to the higher altitude, while panels and (c), (d), show results for the lower altitude. Panels (b) and (d) focus on the sub-micrometer details of the full size distributions plotted in panels (a) and (c).**