# Peer review of "Post-flight analysis of detailed size distributions of warm cloud droplets, as determined in situ by cloud and aerosol spectrometers"

_Atmospheric Measurement Techniques, 2021_

## Referee Comment (RC1)

This manuscript deals with the analysis of cloud droplet size spectra measured by a commercial probe, the Cloud and Aerosol Spectrometer (CAS.) Topics include the error caused by the "ripples" in the Mie program that predicts the amount of forward-scattered light in the probe for each droplet and its location in the size bin related to that amount of scatter. Dealing with the error caused by the "ripples" is described. The distribution of size bins for field use is shown and compared to other size-bin options that appears to be the main topic of the paper. In particular, the authors note the possibility of using PbP (full particle by particle data) CAS files that record the scatter for each individual droplet. They show how this high resolution PbP data creates much greater detail than the field size bins. A final topic describes the effect of contaminated droplets on the output of the CAS.

This paper is a useful contribution for CAS users, especially if they wish to deal with the added complexity of using finer-scale size bins in describing droplet size spectra. While this is a well-written paper, some additional information is recommended to be included for making the contents more suitable for potential users.

Comments:

1.      We know that droplet spectrometers, such as the CAS, have a very small sensitive volume where the forward scattered light of droplets is measured. This significantly limits their ability to measure droplet size spectra over short in-loud distances. The authors note the droplet spectra is "conveniently" measured in-cloud every "sampling instance" of1s, which at a typical research-aircraft velocity of 100 m/s is equivalent to 100m. The total sample volume seen by the CAS over this distance appears to give enough droplets ("thousands") for the CAS to sample, giving meaningful spectra for "normal clouds" as noted by the authors. Some spectrometer data has been published at 10 Hz, but here the ambient droplet concentration must be quite large for meaningful results. Given that ambient droplets can be described as being approximately distributed randomly in space means that the CAS must measure enough droplets to achieve acceptable statistical uncertainty in the resulting spectra. For that reason, obtaining CAS spectra over distances < 10m at the given aircraft velocity usually appears unrealistic.

        The preceding comments apply directly to PdP data files that the authors describe as containing scattered light (and time arrival data) data for each of the first 292 droplets for each "sampling instance" of 1s. Clearly, such a low number of droplets for each 1s will not yield much useful information on desired high-resolution size spectra, therefore PdP from multiple 1-s "sampling instances" must be combined to achieve statistical significance. In the authors' given spectral plots this is apparently done because of the flight duration of the measurements covering "several minutes".

        The authors state that their effort included a "…complicated and time-consuming analysis of the PbP files…". Thus, their PbP approach may be of limited practicality; although, it still may be useful if a limited number of high-resolution spectra are needed corresponding to lengthy sampling periods.

        In addition to showing the spectra from the authors' measurements over several minutes, it would be useful for the reader if the authors also included spectral data for a 1-s interval and its associated companion PbP spectra, given that droplet spectra are often presented for this time interval. An estimate of how many intervals must be combined to obtain useful high-resolution PbP spectra would also be useful.

2.      Droplet spectra from spectrometers are often given as continuous data without error bars. Since cloud droplets are approximately distributed randomly in space (~ Poisson distribution) it is possible to estimate from the droplets' count in each size bin the uncertainty of the count. This is rarely ever done. Please discuss how this statistical uncertainty affects the accuracy of your spectra measurements, especially those associated with PbP.

3.      In Fig. 6 the authors show the effect of a 10% error in the scattered light measurement on the droplet spectra. This amount of error is less than the error spec provided by the CAS manufacturer as noted by the authors. Figure 7 shows a relative error of 20% for the flight-line sample volume, and shows the expected

result on cloud parameters when the scattered-light error increases to large values. Are the abscissa error values realistic? Given these substantial errors, does the detailed analysis of the Mie "ripples" in the paper lead to errors of the same magnitude, or can they be ignored in comparison? Please comment.

4.      Section 6 uses Mie theory to estimate the effect of different refractive indexes on various droplet properties. For example, Fig. 11 illustrates the obvious strong effect on droplet properties for small values of the imaginary index. To put this result into perspective, can the authors indicate where typical ambient cloud droplets fit into the 3-D plots of Fig. 11?

Minor Comments:

L52 -   what is meant by "…cast into the generic name…" Suggest removing this part and using '…instrument is the Cloud…'

Fig. 3 -   The numerical values of the ordinate axis of the right-hand LWC plot are incorrect.

---

## Author Comment (AC1)

Answers to RC1 (Anonymous Referee #2, supplementary comments)

The authors are grateful for the Referee's attentive inspection of the text, for the generous appreciation of this work and mostly for the Referee's additional comments and suggestions for further improvement to the paper. In the following, detailed responses to those comments are presented.

**Referee's comment #1**: We know that droplet spectrometers, such as the CAS, have a very small sensitive volume where the forward scattered light of droplets is measured. This significantly limits their ability to measure droplet size spectra over short in-loud distances. The authors note the droplet spectra is "conveniently" measured in-cloud every "sampling instance" of 1s, which at a typical research-aircraft velocity of 100 m/s is equivalent to 100m. The total sample volume seen by the CAS over this distance appears to give enough droplets ("thousands") for the CAS to sample, giving meaningful spectra for "normal clouds" as noted by the authors. Some spectrometer data has been published at 10 Hz, but here the ambient droplet concentration must be quite large for meaningful results. Given that ambient droplets can be described as being approximately distributed randomly in space means that the CAS must measure enough droplets to achieve acceptable statistical uncertainty in the resulting spectra. For that reason, obtaining CAS spectra over distances < 10m at the given aircraft velocity usually appears unrealistic.

The preceding comments apply directly to PdP data files that the authors describe as containing scattered light (and time arrival data) data for each of the first 292 droplets for each "sampling instance" of 1s. Clearly, such a low number of droplets for each 1s will not yield much useful information on desired high-resolution size spectra, therefore PdP from multiple 1-s "sampling instances" must be combined to achieve statistical significance. In the authors' given spectral plots this is apparently done because of the flight duration of the measurements covering "several minutes".

The authors state that their effort included a "…complicated and time-consuming analysis of the PbP files…". Thus, their PbP approach may be of limited practicality; although, it still may be useful if a limited number of high-resolution spectra are needed corresponding to lengthy sampling periods.

In addition to showing the spectra from the authors' measurements over several minutes, it would be useful for the reader if the authors also included spectral data for a 1-s interval and its associated companion PbP spectra, given that droplet spectra are often

presented for this time interval. An estimate of how many intervals must be combined to obtain useful high-resolution PbP spectra would also be useful.

**Answer to comment #1**: As the Referee correctly pointed out, the PbP data becomes statistically significant for relatively large sequences of one-second sampling instances. This observation has been used to construct detailed dimensional distributions of droplets during flight lines which typically take "several minutes" each. The methodology described in the paper has been designed for dealing with such large cloud sections. Analysing the data for each one-second sampling instance would eventually bring information on spatial fluctuations of cloud properties, a matter which is beyond the purpose of our study. Moreover, with the highly improbable exception of a very spatially homogeneous cloud, such one-second spectral data is more likely statistically incompatible with the distribution of the ***first*** ~ 290 particles detected in the sampling instance, which are stored in the PbP file. These particles are not randomly picked from the set detected in the whole sampling instance, they are only the first in this set and one may suspect that this very choice might induce a statistical bias. However, when combining many consecutive sampling instances, one can expect that the statistical bias (if any) is effectively eliminated and that the PbP data behaves statistically in a similar way as the bulk data file.

A further related question could be: How many sampling instances are necessary to achieve statistical consistency of the PbP data? The answer, as pointed out by the referee, would clearly depend on the density of the droplet population in the given flight segment. The denser the corresponding cloud area, the longer the recording should be in order to achieve a statistically consistent PbP data file. In general, by assuming that in a sampling instance the instrument can measure enough particles to produce a significant distribution, we conjecture that the total length of the PbP data file should be at least equal to the average number of qualified particles during a single sampling instance.

In order to make those points clearer in the text, the authors have added a new paragraph starting on line 278: *"In this connection, we note that the statistical consistency of the PbP data might be questioned due to the fact that the corresponding particles are not randomly picked from the set detected in the whole sampling instance. They are the first ~ 290 in this set and one may suspect that this very choice might induce a statistical bias. However, when combining many consecutive sampling instances, one can expect that the statistical bias (if any) is effectively eliminated and that the PbP data behaves statistically in a similar way as the bulk data file. A further related question could be: How many sampling instances are*

*necessary to achieve statistical consistency of the PbP data? The answer would clearly depend on the density of the droplet population in the given flight segment. The denser the corresponding cloud area, the longer the recording should be in order to achieve a statistically consistent PbP data file. In general, by assuming that in a sampling instance the instrument can measure enough particles to produce a significant distribution, we conjecture that the total length of the PbP data file should be at least equal to the average number of qualified particles during a single sampling instance."*

Regarding the "…complicated and time-consuming analysis of the PbP files…" that would limit the practicality of the method, we would like to point out that this statement is made at the end of Section 5 and refers only to the **accuracy evaluation** of the cloud microphysical parameters, which is indeed a lengthy (nevertheless automatic) process. The spectral analysis itself, based on the PbP files, is much more straightforward, but is still only doable in the post-flight stage.

**Referee's comment #2**: Droplet spectra from spectrometers are often given as continuous data without error bars. Since cloud droplets are approximately distributed randomly in space (~ Poisson distribution) it is possible to estimate from the droplets' count in each size bin the uncertainty of the count. This is rarely ever done. Please discuss how this statistical uncertainty affects the accuracy of your spectra measurements, especially those associated with PbP.

**Answer to comment #2**: The authors assert that error bars are not a proper way to express uncertainties related to distributions. An error bar can be used for indicating the local imprecision of a certain quantity whose values are pointwise independent. For example, in the emission spectrum of a certain sample expressed as a function of the wavelength, each measured intensity fits into a certain precision interval, which usually depends on the wavelength, and can be shown graphically as an error bar. Otherwise, the values taken by the intensity at different wavelengths are independent. The situation is different for statistical distributions (for which reason it would be recommended to avoid terms as "dimensional spectrum" and replace them with "dimensional distributions") where, due to the normalization requirement (there is always a **given number** of objects to be distributed over a certain range of a variable), a variation at a certain point determines a change in **the diagram as a whole**. For this reason, we consider that the error bars are not suited (they can actually be misleading!) for illustrating the imprecision of a statistical distribution and this is a possible reason for which they are not normally used in the literature.

However, to compensate for this deficiency and to attempt a graphical illustration for the imprecision of a size distribution, the authors constructed Figure 6 in the manuscript, where the "nominal" and "distorted" (due to the errors induced by the proposed method) versions of a size distribution have been shown on the same diagram. Obviously, similar distortions can be observed in size distributions over any structure of dimensional bins. The choice of 50 equal bins for Figure 6 was made mainly for reasons of graphical relevance.

**Referee's comment #3**: In Fig. 6 the authors show the effect of a 10% error in the scattered light measurement on the droplet spectra. This amount of error is less than the error spec provided by the CAS manufacturer as noted by the authors. Figure 7 shows a relative error of 20% for the flight-line sample volume, and shows the expected result on cloud parameters when the scattered-light error increases to large values. Are the abscissa error values realistic? Given these substantial errors, does the detailed analysis of the Mie "ripples" in the paper lead to errors of the same magnitude, or can they be ignored in comparison? Please comment.

**Answer to comment #3**: Thank you for pointing this out. The diagrams of Figure 7 show some kinds of lookup diagrams where the errors in the measured scattered light determine the accuracy of the methodology proposed in the paper (or, in the Referee's words, the detailed analysis of the Mie "ripples"). As stated in the Introduction (lines 64-67), the paper is not attempting to deal with the CAS errors in the scattered light measurements. Various manufacturers give different values for these errors (which have also a strong variation with the level of the intensity of the scattered light). The scattered light measurement errors are expected to decrease with time, when more precise instruments will become available commercially. The values on the abscissa of Figure 7 are therefore mainly generic. The diagrams can be of interest if a certain CAS user knows the errors with which the instrument measures the scattered light on its different amplification stages. The user can then simply read the ordinate indications for the final errors of the various cloud parameters. Of course, the diagrams have been constructed for a certain value of the sample volume, which is another (essentially functional) parameter of the instrument whose knowledge needs to be continuously improved. The error diagrams should be reconstructed for every level of error of the sample volume.

To improve the discussion, we have added: *"It is also remarkable that, while the generic range of the relative errors of the FWSCS measurements is relatively wide, some cloud parameters (like the extinction coefficient or the effective diameter) tend to be determined with better final*

*accuracy through this methodology (at least over some ranges) than that provided for the measured FWSCS values."* on line 485.

**Referee's comment #4**: Section 6 uses Mie theory to estimate the effect of different refractive indexes on various droplet properties. For example, Fig. 11 illustrates the obvious strong effect on droplet properties for small values of the imaginary index. To put this result into perspective, can the authors indicate where typical ambient cloud droplets fit into the 3-D plots of Fig. 11?

**Answer to comment #4**: It was the initial authors' intention to look for some regularity in the optical properties of contaminated cloud droplets. If their complex refractive index would have depended only on the nature of the incorporated aerosol particles, then there could have been chances to find some typical behaviour. Unfortunately, the variability of the droplets' optical properties equally depends on the amount of "ingested" aerosol, which is strongly correlated to the aerosol load in the atmosphere at various altitudes. These factors make the definition of "typical ambient cloud droplets" very difficult, at least to the authors' knowledge.

Instead, we propose in the manuscript (lines 623-628) that the (average) optical properties of the cloud droplets measured in a flight line be estimated through post-flight optical analysis of collected samples of cloud droplets/cloud water during that flight line.

**Referee's minor comments**:

**L52** - what is meant by "…cast into the generic name…" Suggest removing this part and using '…instrument is the Cloud…'

**Answer**: The correction suggested by the Referee has been performed.

**Fig. 3 -** The numerical values of the ordinate axis of the right-hand LWC plot are incorrect.

**Answer**: The number of decimal digits in the values on the ordinate axis has been set to 1, by mistake. The error has been corrected in the new version of the text.

---

## Author Comment (AC2)

Answers to RC2 (Darrel Baumgardner)

The authors would like to thank Dr. Darrel Baumgardner (RC2) for taking his time to review the submitted manuscript, especially considering his overwhelming expertise and well-known prestige in the field of optical particle counters. We have now updated the manuscript text to address his concerns and hope that the detailed point-by-point responses presented below will convince him to reconsider his initial recommendation to withdrawal the manuscript. All line numbers in the responses refer to the revised version of the manuscript.

**Comment #1**: The interest of the authors in using the particle by particle data (PbP) is a worthy objective; however, they have overlooked a number of critical factors in their methodology development that puts in question the usefulness of the smoothing technique until they address these factors. Before embarking on this development and writing of this manuscript they should have contacted us at Droplet Measurement Technologies and discussed what they planned to do. This would have possibly clarified for them why their approach needed to be reviewed and modified.

**Answer to comment #1**: Thank you for acknowledging the effort involved in using the PbP data to eventually improve the precision in sizing with the CAS. We agree that some of the methods were unclear in the submitted manuscript. We have now taken the chance to add more complete explanations in several parts of the text that, hopefully, will make our arguments clearer.

In terms of contacting DMT, we would like to point out that we (SNV, AC and VF) had several rounds of discussions with DMT scientists and product engineers about several of the "critical factors in our methodology". DMT product engineers eventually informed us of some specific values of our instrument's functional parameters such as that of the angular interval where the scattered light is collected (4.0° – 13.5°).

**Comment #2**: Secondly, they have overlooked a number of important publications that have already explored the issues that they discuss and addressed how to account for the ambiguities in size and scattering cross section. I have listed these below. Although several refer to the FSSP and not the CAS, the collection angles are similar and measurement principals are the same.

**Answer to comment #2**: Thank you for pointing these studies out. We completely agree that we should have included them and have now done so in the revised version of the manuscript as follows: Brenguier et al. (1998), on lines 164, 194 and 415; Granados-Muñoz et al. (2016), on lines 121 and 194; Pinnick et al (1981), on lines 163 and 190; Rosenberg et al. (2012), on lines 120, 164 and 347. However, it is worth noting that the article by Rosenberg et al. (2012) was actually cited twice in the Discussion version of the paper. After looking more closely into previous approaches of the possible influence of various sizing errors on the ensuing droplet distributions, we have also included: Dye and Baumgardner (1984), on lines 120 and 190, Cooper (1981), on line 414 and Baumgardner et al. (2017), on line 139, to the manuscript.

**Comment #3**: Thirdly, you have to take into account two important factors when carrying out the Mie calculations: 1) the droplets are being illuminated by a laser whose intensity cross section is not precisely uniform, which means that the high resolution oscillations are smoothed out (the authors state on line 157, "In older descriptions of the forward scattering spectrometers (originally used for aerosol sizing measurements, see Baumgardner et al., 1992) this aspect seemed to be overlooked and some smoothened versions of the FWSCS vs. diameter diagrams appeared to have been used.", but we were well aware of the oscillation but took into account the multimodal aspects of the FSSP lasers. And 2) the authors need a better understanding of how the scattering angles are obtained and understand that they are not a precise 4-13.5. Why? Because the scattering angles are determined by the distance of the measured droplet from the dump spot, the diameter of the dump spot and the diameter of the aperture. This distance varies because the depth of field is of finite width. This means that the positioning of the peaks and valleys in the FWSCS shift slight, smearing out the fine detail that the authors show in their figures. This has to be taken into account.

**Answer to comment #3**: First of all, the authors totally agree that the phrase indicated by the Referee is inappropriate and would like to apologize for misrepresenting the study. It actually dates from the earliest of the many versions of the text and is now removed in the revised manuscript and changed to *"This aspect, which generates sizing ambiguities through FWSCS measurements, has been known and analysed for a long time in the literature (Pinnick et al., 1981; Baumgardner et al., 1992; Brenguier et al, 1998)"* on line 163.
Regarding the "Mie calculations", which we understand as obtaining the particle size distribution through a comparison with the exact FWSCS-diameter diagram, the authors

believe that there are two ways to deal with this. Firstly, the FWSCS-diameter diagram (or Mie diagram) can be constructed by embedding in it all the instrumental errors (including those resulting from the cross-sectional non-uniformity of the laser beam and the imprecise knowledge of the angular interval for collecting the scattered light). By accounting for all of these errors, the fine details of the Mie diagram will certainly be smeared out (as the referee states) to a smoother curve, which allows for the establishment of a relatively small set of uneven (and unequivocal) size bins over which the size distribution can be built up. This is the typical approach that allows fast practical evaluations of the in-flight data. A related comment has been added in the new version of the manuscript on line 189: *"For practical purposes, partitions of the diameters' range in uneven size bins have been previously proposed in the literature (Pinnick et al., 1981; Dye and Baumgardner, 1984). Such procedures are actually fitting of the exact FWSCS-diameter diagram with a discrete monotonic plot of response thresholds, each corresponding to a size bin limit. The fitting should be made so that the differences between the threshold plot and the exact Mie diagram can be assimilated to the resultant of the various errors generated in the FWSCS measurement process (Brenguier et al., 1998; Granados-Muñoz et al. 2016)."*

Alternatively, starting from the finest, "exact" form of the Mie diagram, one can take each measured FWSCS value and, in case it is equivocal, equally "distribute" it to all its intersections with the Mie diagram. That is to say, instead of counting one particle for one of the intersections, we count a fraction for each intersection in the size distribution. For example, if there are 5 intersections for a certain measured FWSCS value, at the diameter values d1, d2, ..., d5, consider in the distribution that, for each d1, d2, ..., d5, we have 1/5 of a particle. Then, to account for the measurement errors, which are assumed as known for each value of the FWSCS, compute the shift it produces in the distribution and the ensuing errors of other cloud parameters that result from various averages. This proposed approach, which is briefly described here, is discussed in detail within Section 4 of the manuscript. The difference between the two methods has also been outlined in two new paragraphs added in the revised version of the manuscript, starting on line 413: *"At this point, it should be mentioned that the influence of the sizing errors on the resulting droplet distributions has been previously addressed in detail in the literature (Cooper, 1981; Baumgardner et al., 1992; Brenguier et al., 1998) through an ingenious mathematical method based on a transfer matrix that takes the measured distribution into the actual one. The elements of the matrix are actually probabilities that a certain measured particle of a given diameter be counted in a different size bin. The transfer matrix has to be constructed for each instrument and its elements embed both the*

*errors generated by the FWSCS measurements and those ensuing from the ambiguities in the comparison with the Mie diagram.*

*The present study tries a different approach, by separating the measurement errors of the FWSCS values (which stem from various hardware issues and have to be known) and by considering in greater detail the uncertainties generated by the comparison with the "exact" Mie diagram."*

In the end, both methods basically lead to the same macroscopic result, but each one has its specific advantages. The first method is simpler, fast and suitable for rapid in-flight analyses. The second method is more computationally demanding and can be applied only post-flight and only if one knows the measured values for individual particles, which means the availability of the PbP files. As for the advantages of the second and proposed method, they all stem from the possibility of obtaining fine size distributions which may provide the cloud composition in greater detail and also could be prospectively useful in operating future, likely more precise versions of optical particle counters. Moreover, as exemplified in Figures 3 and 4 of the manuscript, the fine size distributions can be readily used to construct distributions over coarser structures of (even or uneven) size bins (one can actually reproduce – and we did this as a test of consistency – the size distributions obtained with the first method). As pointed out on line 468, coarser size distributions are more convenient for evaluating the effect of measurement errors.

**Comment #4**: Finally, although I would really like to see the type of fine detail in the size distributions that the authors show in Figs. 3 and 4, and attribute to natural microphysical features, I suggest that they look carefully where those peaks and valleys fall in the size distribution and then take a careful look at their FWSCS diagrams and they will see that many, if not most, of these feature are a result of the Mie ambiguities. This is why they have to read Brenguier et al who actually uses those features to do quality checking of their FSSP.

**Answer to comment #4**: The Referee suggests that the features appearing in the size distributions result from the Mie ambiguities. The authors cannot agree with this statement. The ambiguities of the Mie diagram cannot play a role in the appearance of various features in the size distribution constructed through the proposed methodology for the following straightforward reason: when a particle is detected with an ambiguous FWSCS value, the counter does not record several particles but several "fractions" of a single particle (as discussed on lines 250-255). So, only if there are more particles with some FWSCS value, there

will be a peak in the distribution in the size range where the given FWSCS value intersects the Mie diagram. If there are few or no particles with that value of the FWSCS, then there will be a minimum in the distribution. If the maxima or minima of the detailed size distributions would be "a result the Mie ambiguities" then all the detailed size distributions obtained through the proposed methodology should have the same appearance. However, it is sufficient to look, for example, to panels (a) and (c) of Figure 12 to observe that the detailed size distributions constructed for two flight lines show pretty different features, even if they were obtained with the same ("exact") Mie diagram.

**Comment #5**: Line1 67: The scattering cross section is not quasi-monotonic. It oscillates.

**Answer to comment #5**: The expression "quasi-monotonic" has been replaced by "oscillatory" on line 69 in the present revised version of the manuscript.

**References**:

Baumgardner, D., Dye, J. E., Gandrud, B. W., and Knollenberg, R. G.: Interpretation of measurements made by the forward scattering spectrometer probe (FSSP-300) during the airborne arctic stratospheric expedition, Journal of Geophysical Research, 97(D8), 8035-8046, https://doi.org/10.1029/91JD02728, 1992.

Baumgardner, D., Abel, S. J., Axisa, D., Cotton, R., Crosier, J., Field, P., Gurganus, C., Heymsfield, A., Korolev, A., Krämer, M., Lawson, P., McFarquhar, G., Ulanowski, Z., and Um., J.: Cloud Ice Properties: In Situ Measurement Challenges, in: Meteorological Monographs, 58(1), American Meteorological Society, 9.1-9.23, https://doi.org/10.1175/AMSMONOGRAPHS-D-16-0011.1, 2017.

Brenguier, J.-L., Bourrianne, T., De Araujo Coelho, A., Isbert, J., Peytavi, R., Trevarin, D., Weschler, P.: Improvements of Droplet Size Distribution Measurements with the Fast-FSSP (Forward Scattering Spectrometer Probe), Journal of Atmospheric and Oceanic Technology, 15, 1077-1090, https://doi.org/10.1175/1520-0426(1998)015<1077:IODSDM>2.0.CO;2, 1998.

Cooper, W. A.: Effects of coincidence on measurements with a Forward Scattering Spectrometer Probe. J. Atmos. Oceanic Technol., 5, 823–832, https://doi.org/10.1175/1520-0426(1988)005<0823:EOCOMW>2.0.CO;2, 1988.

Dye, J. E. and Baumgardner, D.: Evaluation of the Forward Scattering Spectrometer Probe. Part I: Electronic and Optical Studies, J. of Atmos. Oceanic Technol., 1(4), 329-344, https://doi.org/10.1175/1520-0426(1984)001<0329:EOTFSS>2.0.CO;2, 1984.

Granados-Muñoz, M. J., Bravo-Aranda, J. A., Baumgardner, D., Guerrero-Rascado, J. L., Pérez-Ramírez, D., Navas-Guzmán, F., Veselovskii, I., Lyamani, H., Valenzuela, A., Olmo, F. J., Titos, G., Andrey, J., Chaikovsky, A., Dubovik, O., Gil-Ojeda, M., and Alados-Arboledas, L.: A comparative study of aerosol microphysical properties retrieved from ground-based remote sensing and aircraft in situ measurements during a Saharan dust event, Atmos. Meas. Tech., 9, 1113–1133, https://doi.org/10.5194/amt-9-1113-2016, 2016.

Liu, B. Y. H., Berglund, R. N., and Agarwal, H. K.: Experimental Studies of Optical Particle Counters, Atmos. Environ., 8(7), 717–732, https://doi.org/10.1016/0004-6981(74)90163-2, 1974.

Pinnick, R. G., Garvey, D. M., and Duncan, L. D.: Calibration of Knollenberg FSSP Light Scattering Counters for Measurement of Cloud Droplets, J. Appl. Meteor, 20, 1049-1057, https://doi.org/10.1175/1520-0450(1981)020<1049:COKFLS>2.0.CO;2, 1981.

Rosenberg, P. D., Dean, A. R., Williams, P. I., Dorsey, J. R., Minikin, A., Pickering, M. A., and Petzold, A.: Particle sizing calibration with refractive index correction for light scattering optical particle counters and impacts upon PCASP and CDP data collected during the Fennec campaign, Atmos. Meas. Tech., 5, 1147–1163, https://doi.org/10.1002/qj.777, 2012.